# Element Enterprise Tycoon: Playing Board Games to Learn Chemistry in Daily Life

**Jen-Che Tsai [1], Shih-Yeh Chen [2], Chun-Yen Chang [1,\*] and Shiang-Yao Liu [1,\*]**

[1] Graduate Institute of Science Education, National Taiwan Normal University, Taipei 11677, Taiwan; lansk110792@gmail.com

[2] Taichung Municipal Dali High School, Taichung 41260, Taiwan; 80345001s@gmail.com

[*] Correspondence: changcy@ntnu.edu.tw (C.-Y.C.); liusy@ntnu.edu.tw (S.-Y.L.); Tel.: +886-27734-6751(C.-Y.C.); Tel.: +886-27734-6807(S.-Y.L.)

**Abstract:** This article reports the design of a scientific board game, named "Element Enterprise Tycoon" (EET), which creates a scenario combining chemical elements, techniques, and products in daily life. The game cards are designed to motivate students not only to retrieve information about chemical elements, but also to be proficient in chemistry. Moreover, the game creates opportunities for group interactions and competitions to engage students in learning chemical elements as they do in regular science curricula. The EET has been field-tested with a group of middle school students to evaluate its applicability. Empirical data show that students improve their understanding of chemistry concepts with a median level of effect size. In particular, students achieve better performance in terms of chemistry-related technique concepts. The follow-up interviews reflect students' positive feedback and attitudes toward science learning through board game playing and their willingness to continue to play the game. It is suggested that learning through science games can indeed help students learn new chemical knowledge.

**Keywords:** board game; chemistry elements; game-based learning

## 1. Introduction

The use of scientific board games for teaching and the gamification of scientific concepts has become an emerging teaching trend [1]. Several board games for supporting the teaching of chemistry courses have been employed by many schoolteachers and science education scholars [1–14]. Research shows the difficulties in helping students achieve higher learning motivation by using only the traditional modes of memorization and repeated practice, which may cause students to lose their interest in science as well as their willingness to further study and explore science independently [2,3]. It is evident that using board games for teaching can improve students' motivation for learning, reduce the learning difficulty of complex concepts, and subsequently improve their learning efficiency [4–7]. Teaching through board games not only enhances students' study of scientific knowledge but also develops their scientific competencies, such as their problem-solving, collaboration, communication, and negotiation capabilities [8,9].

Martí-Centelles and Rubio-Magnieto [1] consider the periodic table of chemical elements to be the basis of chemistry and the most important scientific language. In the past, teachers often taught the periodic table of chemical elements by requiring students to memorize, recite, and learn through rote formulas. Although students may have been able to write down correct answers for exams, they failed to understand the application of chemical elements and the connection between chemical elements and their daily lives. Franco-Mariscal et al. [3] introduced a chemical element board game in the form of a card collection game (chemical family). The main purpose of this game was to help

students understand and memorize the "chemical family" of chemical elements and their use. Bayir [5] used a question-and-answer guessing game about the periodic table of chemical elements to help students understand the physical and chemical properties of different chemical elements. The contents of the game cards included atomic number, orbital chemistry, color, and chemical family. Martí-Centelles and Rubio-Magnieto [1] launched a board game concerning the periodic table, which was based on a commercial version of the card game UNO. Their game allowed students to understand different chemical families and the ordering of chemical elements.

The conversion of chemical knowledge into board games may improve students' motivation and concentration. However, if the board games overemphasize the learning of scientific knowledge itself, the connections and gaps between chemical elements and students' daily lives might be ignored. Previous research suggested that science board games should simulate students to apply scientific knowledge when dealing with life situations. Moreover, if the only topic that was gamified was the scientific concept, for middle school students who have studied only the basic concepts of chemistry, it is difficult to maintain interest in playing the games [15–17]. Therefore, there is still room for improvement in the current scientific board games in teaching the periodic table of the chemical elements. This study posits that a useful educational board game should encourage students' interest in chemical concepts, stimulate the integration of the game process and students' real-life situations, and produce a learning transfer, encouraging students to apply what they have learned to their everyday lives.

## 2. Research Purposes

This study has designed a scientific board game, named "Element Enterprise Tycoon" (EET), which intends to help beginning learners understand the concept of chemical elements and to strengthen their understanding of the connections and applications of chemical elements in real-life situations. At present, most of the board game research on the periodic table discussed in the previous section focuses on high-school and college students (grade 10 and beyond) [3,8,9,11]. Very few studies focus on middle school students [5]. According to science curriculum guidelines in Taiwan, the periodicity of chemical elements is a new scientific topic for middle school students to learn (grade 8). Most teachers require their students to memorize and recite the periodic table according to the atomic sequence or the ordering of chemical families. However, with this approach, students only remember the names and properties of certain chemical elements but do not recognize the daily applications and techniques associated with those chemical elements. This teaching method is still a common way to support satisfactory academic performance in students, but may also make students feel stressed and less interested in the learning of chemical elements because it results in a clear gap between the scientific concept of the periodic table of chemical elements and students' actual life experience.

The research focus is also on testing the applicability of the EET board game for middle school students to learn the periodic table of chemical elements. Based on a total of 38 chemical elements commonly used in life and industry, as well as relevant techniques and products for real life, the learning of the periodic table of chemical elements is combined with students' actual life experience. The board game should be field-tested with a group of students to ensure the outcomes. To evaluate students' understanding of the game mechanism and chemical concepts through the playing of the board game, group interviews and conceptual tests are used to determine whether these students understood the scientific concepts that the game designer intended to convey, as well as the operational mechanisms during their actual play.

## 3. Game Content and Rules

In this study, based on the literature review, the chemical concepts in board games may be divided into three categories. The first category is organic chemistry-related topics, such as the nomenclature of organic compounds, organic synthesis, chemical structure, functional groups, chemical reaction formulas, and other chemical concepts [4,7–13]. The second category includes topics concerning the periodic table, such as the atomic order of chemical elements, atomic weight,

melting point, boiling point, color, and the use of chemical elements [1,3,5,14]. The last category refers to other chemistry-related topics, such as the understanding of the experimental apparatuses that are commonly seen and used in chemical laboratories [6], as well as chemical elements in the natural cycle [14]. Among them, the periodic table is a very important part of chemistry education and is the main subject of the current research.

Figure 1 briefly presents the process of the development of the EET board game. Step 1 is to find out students' difficulties in learning chemical elements and the periodic table. Students often have low learning motivation because they do not know how chemical elements are relevant to their life experiences [15,16,18]. Therefore, in step 2, the researchers have reasons to decide which chemical elements and relevant information about their techniques and products should be selected for inclusion in the card game. Step 3 is to design the game mechanism, including the rules and winning goals. In step 4, the researchers conducted internal evaluations through a penal review of the game contents and experiencing pilot gameplay. Some parameters have been adjusted to ensure the applicability of the board game.

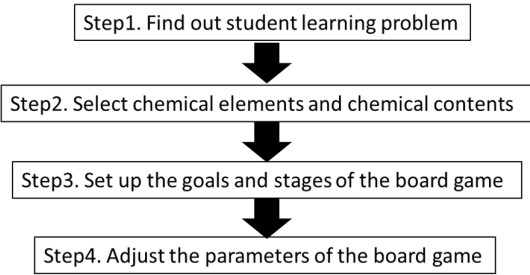

**Figure 1.** Steps for developing the board game**.**

This board game is a typical card game where players collect points through card collection. The game scenario is to assign each student as the president of a company who is responsible for collecting different chemical elements so that she or he can develop new products. The goal of the game is to win the most points or have the highest score. Students may acquire financial gains by selling their products or obtain the cards they desire to possess. EET has four main types of cards: element cards (a total of 38 chemical elements), technique cards (a total of 36 techniques), product cards (a total of 55 products), and opportunity cards (a total of three kinds: money cards, actions cards, and sales cards). Students may wish to combine the chemical elements with related techniques and applications by forming a set of cards that includes the element, technique, and product cards. The more sets of cards the player has, the higher the score. As they collect the cards, students are expected to understand the relationships between chemical elements, related techniques, and applied products. Students can use the action card to obtain the cards they desire from the other players. They can also obtain a variety of sets of cards through attacking or defending activities to reach the goal of the game.

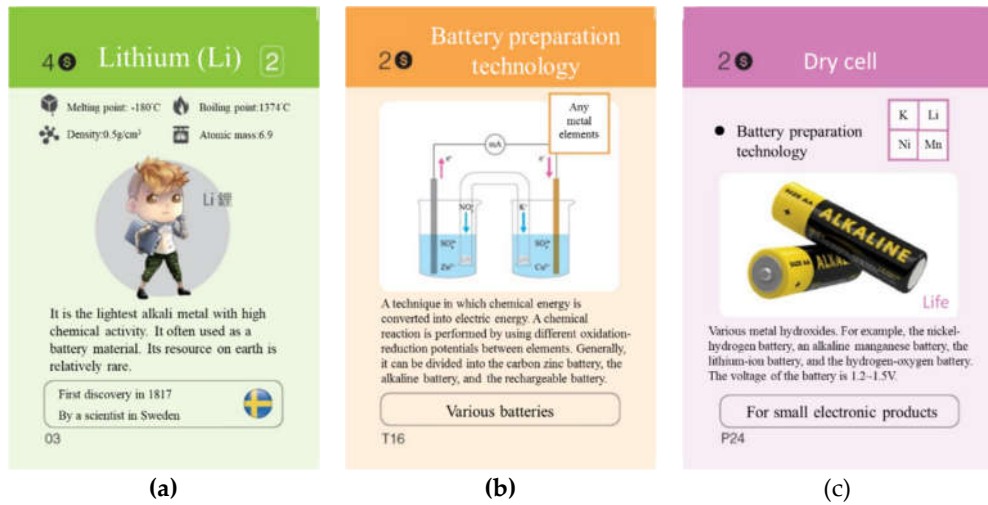

**Figure 2.** Examples of cards: (a) Element cards; (b) Technique cards; (c) Product cards.

### 3.1. Element Cards

Element cards are the most basic cards in the game. Those students without element cards will not be able to develop new techniques or products. The elements on these cards contain the first 30 elements of the periodic table: hydrogen (H), helium (He), lithium (Li), beryllium (Be), boron (B), carbon (C), nitrogen (N), oxygen (O), fluorine (F), neon (Ne), sodium (Na), magnesium (Mg), aluminum (Al), silicon (Si), phosphorus (P), sulfur (S), chlorine (Cl), argon (Ar), potassium (K), calcium (Ca), scandium (Sc), titanium (Ti), vanadium (V), chromium (Cr), manganese (Mn), iron (Fe), cobalt (Co), nickel (Ni), copper (Cu), and zinc (Zn). EET also includes metallic and nonmetallic elements related to applied products that are often used in daily life, such as arsenic (As), silver (Ag), cadmium (Cd), tin (Sn), iodine (I), tungsten (W), gold (Au), and uranium (U). There is a total of 38 elements in this category. Element cards' information should include element names, atomic order, atomic weight, melting point, boiling point, density, element color, element characterization, discovery date, country of discovery, etc. (shown in Figure 1(a)). There is an upper limit on the number of cards per element, which is determined by the stock of this element in the earth's crust and the atmosphere. For example, there are only two cards for lithium (Li) (the number is shown in Figure 2(a) in the upper right corner). These cards can be used not only in the course of the board game but also in general classroom teaching.

### 3.2. Technique Cards

Technique cards must be researched/redeemed according to the requirements of different elements. Each technique card will indicate the elemental conditions required for development (shown in Figure 2(b)). These cards enable students to establish a connection between chemical elements and technique during the game. This group includes a total of 36 techniques, such as a food processing technique, semiconductor fabrication technique, synthesis reaction, an extraction/purification technique, and nuclear power generation. Technique cards' information includes the name and a scientific introduction to a certain technique, along with the types of products with which this technique is associated. During the process of playing the game, a technique card may be connected to an element card and a product card to form a set of cards or a card group, enabling students to connect the element, technique, and product.

### 3.3. Product Cards

Product cards must also be researched/redeemed according to the needs of different elements. Each product card will indicate the elemental conditions required for research and development (shown in Figure 2(c)). These cards enable students to establish a link between chemical elements and

products during gameplay. Product cards include a total of 55 products, such as food, electronic products, cookware, batteries, medicine, energy, transportation, chemical raw materials, and power plants. Product cards information should include the product name, product description, and application method. The product card must have a corresponding technique card and element card to form a complete card set or group. If students do not understand the combination required for a certain product, they need to read its card information to understand.

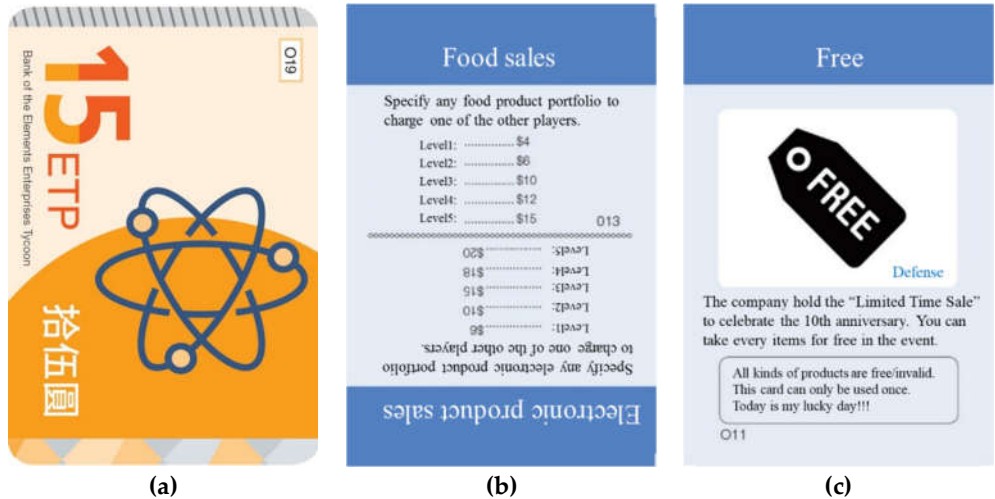

**Figure 3.** Opportunity cards, from left to right: (a) Money card; (b) Sales cards; (c) Actions cards.

### 3.4. Opportunity Cards

Opportunity cards can increase the fun and uncertainty of the game, thus increasing the playability and freedom of the game. There are three types of opportunity cards: money card, sales cards, and actions cards. The money card (shown in Figure 3(a)) is not the key to determining the outcome in the game. It produces a defensive effect against the cost of product sales. The sales card (shown in Figure 3(b)) is used to collect money from other players through the determination of the sale price based on the value of the product that has been developed. If the other player experiences a financial deficit, they may choose to provide collateral. The property on their desktop will be held as collateral by the player who uses the card, resulting in a reduction in points for the financially insufficient player. Actions cards (shown in Figure 3(c)) may be used to attack or defend against other players to increase the interaction and stimulation of the game.

### 3.5. Game Stages and Rules

EET's game stages and rules are simple. The goal is to win the most points within a given time limit. The game has only two stages (Figure 4). First, during the business development stage, each player can choose to play 0 to 5 cards from the starting hand. The number of cards played in this stage must be consistent with the number of cards that can be drawn or researched and developed in the second stage. For example, a student plays four cards, then, he/she can research/draw four cards in the next stage (the research stage). In other words, if a student produces a certain number of cards in this stage, he or she will have to draw the same number of cards in the second stage. Students may choose to attack or collect cards, depending on their personal choices and strategies. Second, during the research stage, the number of cards that can be drawn, researched, and developed is determined according to the number of cards that were issued in the first stage. Players may choose to directly draw opportunity cards or to draw technique cards and product cards instead. When students conduct research and development, they must meet the elemental requirements written on the technique or product card to research, develop, and redeem (the elemental requirements are displayed in the box in the upper right corner of each card). Finally, points are calculated based on the combination of element, technique, and product cards on the desktop. The highest scorer wins!

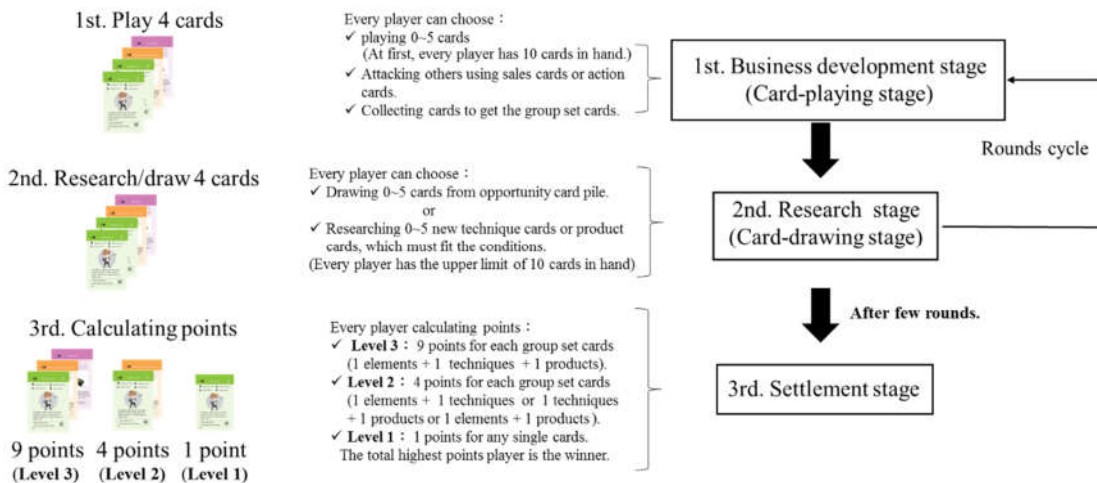

**Figure 4.** Stage descriptions for the board game Element Enterprise Tycoon with an example.

## 4. Methods

The board game was field-tested with 16 students (ages 14–15) from a middle school. These students at this learning stage had learned the basic concepts of the periodic table of chemical elements. The game instruction was implemented to students in six sessions (45 minutes per session, a total of 270 minutes). Specifically, the pre- and post-tests occupied one session each. One session was used for the introduction and a demo of the board game. Two sessions were dedicated to the actual play of the board game. The last session was used to conduct interviews. The teacher does not intervene in the process of student gameplay; it is up to the students themselves to choose their strategy. The teacher's role is only to preside over the game and explain the rules. The assessment tool, containing 20 multiple-choice question items, was compiled by the authors of this study. The questions were designed to assess the scientific concepts of chemical elements and the corresponding technique and product concepts that students possess before and after the game playing. Examples of question items include the following: "Which of the following substances contains silicon?" (element) "Which of the following statements regarding the chemical process or technique to the product is correct?" (technique) and "What is the reason why aluminum windows do not rust easily?" (product). The total possible score of the assessment was 20 points. A semi-structured interview was conducted with six volunteers who accepted the researchers' public invitation. The whole game-playing process was voice and video recorded.

## 5. Results

### 5.1. Learning Outcomes

In this study, 16 students completed the pre- and post-tests. The results (Table 1) show that the students made significant progress in their understanding of scientific concepts, with a median level of effect size (the Cohen's d equaling 0.50). This finding is consistent with the results of other educational board games, which have all achieved satisfying learning outcomes [1,3,5,14]. The study also found that the post-test scores for the technique and product items were all higher than the pre-test scores. Among them, the scores on the technique items showed significant progress ($p = .016$, $d = 1.00$). This result indicates that playing this scientific board game helps students understand technique-related scientific concepts.

**Table 1.** Results of the assessment on the chemistry concepts.

| Assessment Dimensions (# of items) | Mean-pre (range) | SD-pre | Mean-post (range) | SD-post | t value | Effect size |
|---|---|---|---|---|---|---|
| Elements (8) | 4.88 (2–8) | 1.78 | 4.88 (2–8) | 2.06 | 0.00 | 0 |
| Techniques (3) | 0.75 (0–3) | 0.93 | 1.69 (0–3) | 0.94 | 2.70* | 1.00 |
| Products (9) | 4.13 (1–7) | 2.09 | 4.81 (3–7) | 1.27 | 1.48 | 0.39 |
| Total (20) | 9.75 (5–16) | 3.51 | 11.38 (7–16) | 2.98 | 2.26* | 0.50 |

Note: * $p \leq 0.05$.

As to the impact of scientific concepts, those who attended the interview stated which types of cards they thought were most helpful in their learning. Some of their statements are quoted below, with code "B" as male students and "G" as female students.

B1: Technique cards, such as the contact process, which I did not learn before. For the manufacture of sulfuric acid…The isotope separation technique. It is the method for nuclear and thermonuclear weapons, which was not mentioned in class and which I think is of great importance…The school teaches us some rigid information. It is better to teach us some life-related information.

G1: I finally know that fertilizer may be made by the Haber process. The synthesis of nitrogen and hydrogen is carried out during the reaction. I've also realized that there are nitrogen fertilizers, phosphate fertilizers and potassium fertilizers in (commercial) fertilizer, which promote the growth of stems, leaves, and flowers. Fertilizers are (obtained) through chemical processes.

G4: Technique cards, because it teaches you some technical information. For example, what can it produce? What materials are needed? Then, these cards explain how to make and use them.

G3: Product cards for me. They let us know the technique a product needs and how it is made. What technique is needed and what elements are needed.

During the interview, student B1 mentioned that playing this kind of board game helped him gain more scientific knowledge and was more interesting than traditional didactic teaching. Although both student G1 and G4 have been taught the Haber process and its applications in their previous lessons, they were able to better understand its purpose and meaning in real-life situations through the game. Although element items and product items did not show statistically significant progress, they are still helpful for students' learning outcomes. For example, student G3 stated, "It allows us to know a lot of elements. It also teaches us what techniques and what elements are needed to make a product." Compared to common and easy-to-understand element and product items, technique concepts and their links to element and product items are students' primary areas of weakness. Therefore, the results of this game show that students made significant progress in understanding technique-related aspects.

The EET science board game exerted a certain degree of impact on the interaction between students' lives and the society in which they live, which is reflected in students' interview responses. Some of the students' responses are quoted below.

B1: After playing this game, I not only have learned more new things but also have an urge to know even more about them. It makes me want to go online to find out more information.

G1: It lets me know that for everything I buy, it comes with its production method, written on the back of the product…I want to tell my relatives, friends, and even strangers that this board game is very rewarding…I hope that when my family go out and buy something,

they will also read what is written on the back of that product, such as the raw materials…the ingredients and content. They need to read them carefully.

B2: Yes! There is impact. After playing this game…you will know what technique and what elements it needs. This way, you can also teach what you've learned to some of your classmates and friends around you. Playing this board game is worthwhile. It makes you spread the knowledge to others. If I ever see this kind of thing…I will tell them directly about the technique used and the raw materials used.

From the interviews, we can confirm that the gaming mechanism of the scientific board game enables students to connect the element, technique, and product. Students are motivated to learn more about chemical elements (B1) and ingredients in products (G1, B2) and are willing to share what they have learned from the game with others. Students' awareness of the products in their lives has also been raised. For example, after the game, student B1 conducted further research on relevant scientific information, showing his motivation to study after the game. Student G1 now pays more attention to the ingredient list written on the back of a product in her daily life and learns the ingredients in a specific product, raising her awareness of product safety in daily life. Student B2 is willing to take the initiative to share the knowledge he has learned with other students and friends, thus exerting influence on others with his new knowledge. Therefore, the EET board game can help students connect elements, techniques, and products, making them willing to apply the scientific concepts they have learned to the real-life situations they experience. All these findings are consistent with the theory of Antunes et al. [4], according to whom board games allow students to reconstruct their own concepts of knowledge.

## 5.2. Engagement in Board Games

The game mechanisms reflect the degree of students' involvement in the game process. This study uses game mechanisms to link elements, techniques, and products, enabling students to note the scientific information in the game and to generate their own thinking and understanding. The following interview data offer examples of what students think of the mechanisms of the EET board game.

B1: I can understand (the setting of the game mechanism). Mainly it is to look at the substances it produces and the production method. You need to observe and think! In fact, I think the things shown throughout the game are all quite reasonable.

B2: There are some things I have never known before. But with the explanations provided on the card concerning a certain product, I can read and understand that this set of cards can be formed like this…Sometimes I feel that the card combinations of others are weird…I would like him/her to be able to change to a better card. When we make our own combinations, we want to combine some relevant elements so that they look reasonable.

G1: There are some explanations on the product, you can see what technique or element it needs, and then you use the elements to combine and match. You need to see others' cards for a combination. You also need to see if the cards you have drawn are relevant…because you can attack other players and then have your cards assembled smoothly.

As shown by the interviews, students B1 and B2 understand the reasons for the combination of cards. Even if they encounter new information that they have never seen before, they can still understand the reasons for gameplay action from the information provided by the card. Student B1 was able to read the information on the card, take the initiative to think about whether this information matches what has been previously learned, and understand/accept the game process. Students B2 and G1 not only paid attention to their own card combinations but also observed whether other players' card combinations were reasonable. They even attempted to find a way to develop a suitable play strategy to acquire the cards they want and obtain more points. Therefore, a better understanding of the scientific concepts provided by the cards is conducive to students' better

understanding the game's rules and mechanisms. When the students are familiar with the game mechanism, they can achieve the winning goal by collecting sets of cards of different elements, techniques, and products. Li and Tsai [15] believe that game mechanisms may affect the experience of the game process and the learning outcome. Therefore, a real understanding of game mechanisms will help students to learn.

As shown in Figure 5a, students may observe the contents of other players' cards during the game and use strategic thinking and judgment. For example, they may want to hinder other players from collecting a set of cards or to acquire the cards they need from other players. Whichever gaming strategy students choose, they have begun to understand the card information and the connection between elements, techniques, and products. As shown in Figure 5b, they may discover what kinds of cards they need based on the placement of the element, technique, and product. For example, the green cards on the desktop are element cards, the orange ones are technique cards, and the pink ones are product cards. Players form a set of cards by choosing and combining each of the three types of colored cards according to the card information.

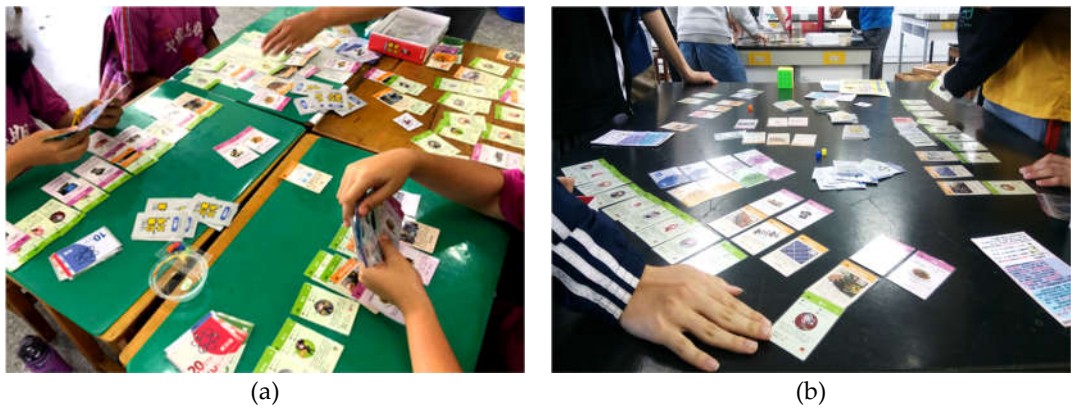

(a)  (b)

**Figure 5.** (a) Students playing the scientific board game Element Enterprise Tycoon. (b) The placement of the kinds of cards.

Finally, we asked students their feelings about playing EET to better understand their thoughts and opinions regarding this scientific board game:

B2: I feel that this game can help you brainstorm and become smarter...it can be combined with real-life situations...I hope I can keep playing this game, because I can learn new knowledge while playing this game. I can think about what my next step is, and I can interact with my opponent.

G3: It helps us to know a lot of elements, what techniques are used in a certain product, and what elements are used. It is nice!

G4: I will be willing to play again. Because there are many kinds of cards and they are open to many different combinations. It renders you a different experience each time you play...It gives you a fresh feeling...You won't feel bored.

Most of the students gave positive feedback during interviews. Student G3 believed that she had learned new knowledge from the game, quenching her thirst for knowledge and satisfying her curiosity. Students B2 and G4 expressed that they would be willing to continue playing the EET scientific board game. They believed that the game helps them think about and learn new knowledge in a more interesting and original way than traditional classroom teaching. The EET scientific board game is clearly popular and highly accepted among these students. Antunes et al. [4] believes that games are an effective way of teaching and that games may arouse students' interest and trigger their persistent devotion in their pursuit of knowledge. The current study also received positive feedback from the students who participated in the EET scientific board game, indicating that the utilization of this game is quite a successful teaching method.

## 6. Conclusions

The EET board game enables students to connect their life experiences and textbook knowledge throughout the course of the game. Through the game mechanism, students are motivated to read and think about the information on the cards and they further gain a complete understanding of a scientific concept by connecting elements, techniques, and products. The results of this study show that students may achieve favorable learning results by playing scientific board games. In addition, through the use of opportunity cards, students are able to broaden their learning horizons by learning different scientific concepts and the connections between elements, techniques, and products through their interaction with others during the game. Therefore, this board game has potential both as a basic chemistry studying method and as supplementary teaching material. The cards in this scientific board game present the course content, enabling the students to observe and learn the chemical knowledge imbued in their daily life through the integration of chemical scenarios/themes and the gaming process itself. It is recommended that schoolteachers endeavor to use scientific board games to assist and enliven their teaching. The effectiveness of a board game embedded in the science curriculum could be assessed by implementing it in a larger group of participants across different learning stages.

**Author Contributions:** Conceptualization, Tsai, J.C.; methodology and formal analysis, Tsai, J.C. and Chen, S.Y.; writing, Tsai, J.C. and Chen, S.Y.; editing and supervision, Chang, C.Y. and Liu, S.Y. All authors have read and agreed to the published version of the manuscript.

**Funding:** This study was financially supported by the Ministry of Science and Technology (MOST) (MOST107-2511-H-003-020-MY2), National Taiwan Normal University Subsidy for Talent Promotion Program and the "Institute for Research Excellence in Learning Sciences" of National Taiwan Normal University (NTNU) from the Featured Areas Research Center Program within the framework of the Higher Education Sprout Project by the Ministry of Education (MOE) in Taiwan.

**Conflicts of Interest:** The authors declare no competing financial interests.

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
