# Peer review of "Element Enterprise Tycoon: Playing Board Games to Learn Chemistry in Daily Life"

_education, doi:10.3390/educsci10030048_

Round 1

Reviewer 1 Report

In principle, the results and contents of the study appear interesting and should be published.

However, some changes should be made in order to ensure comprehensibility for the readers. This begins with a central question: What kind of process is described by the authors here? Is it

a.) about the development of a game (EET) and the subsequent test with pupils

Or

b.) about the test of an already existing game, possibly developed by others?

This should be emphasized more clearly in the abstract and in the introduction, as it seems so far as if EET is a generally known game, but it is not known to this reviewer and probably this appears also for many readers.

Therefore, section 1 Introduction and 2 Research purposes should be revised to make clear to the reader what process is described here. In line 82 the design of EET for pupils is mentioned as the aim of the study. In this respect I assume that EET was developed by the authors within the framework of the study. However, this should then be briefly explained at the beginning of section 3, otherwise this chapter would sound like a description of the game only, but the process of game development would also be of interest to the reader. In this respect, the section from lines 31 to 40 in the introduction also seems to relate more to the design of the game and should possibly be included in chapter 3, as it is inappropriately specific in section 1. As the development of EET is the aim of this work, the timing and interrelation of the different elements of this study should also be clarified. What was the approach? A short explanatory section at the beginning of section 3 would be helpful for the reader's understanding. Another possibility would be a small graphic that relates the development and the survey procedure to each other and illustrates the process.

The abstract should also make clear that EET has been developed and consequently tested. The processuality and the interrelationships in this development should be clearly formulated. So far EET appears as a generally known quantity, here context and explanation for the reader is missing. One or two sentences explaining the basic advantages of games for learning and the development of EET would be very helpful.

Other notes:

The authors sometimes jump between the grammatical tenses, for example in chapter 4 when "were" is used in the first sentence and in the following you use "are". This should be adjusted throughout the article.

Isn't figure 4 actually 2 figures with the same content? Either both photos should be numbered or only one photo should be selected. Another possibility would be again the naming in a and b as in figures 1 and 2.

Author Response

Reviewer 1

Q1. What kind of process is described by the authors here? Is it about the development of a game (EET) and the subsequent test with pupils or about the test of an already existing game, possibly developed by others? This should be emphasized more clearly in the abstract and in the introduction, as it seems so far as if EET is a generally known game, but it is not known to this reviewer and probably this appears also for many readers.

Reply: This article is to report the design of the game (EET) and the field test with a group of pupils to ensure the applicability of the game. We have clarified this by revising the Abstract (highlighted in red) and Research purpose (page 2, line 60-63). There have been developed board game regarding the periodic table of chemical elements, but this is a new board game developed by the authors of this article. The design of this game aims to link chemical elements and chemical techniques and products found in daily life.

Q2. section 1 Introduction and 2 Research purposes should be revised to make clear to the reader what process is described here. In line 82 the design of EET for pupils is mentioned as the aim of the study. In this respect I assume that EET was developed by the authors within the framework of the study. However, this should then be briefly explained at the beginning of section 3, otherwise this chapter would sound like a description of the game only, but the process of game development would also be of interest to the reader.

Reply: A few sentences in the section of Research purposes have been revised to address the efforts of the design and evaluation in this study.

Q3. In this respect, the section from lines 31 to 40 in the introduction also seems to relate more to the design of the game and should possibly be included in chapter 3, as it is inappropriately specific in section 1. As the development of EET is the aim of this work, the timing and interrelation of the different elements of this study should also be clarified. What was the approach? A short explanatory section at the beginning of section 3 would be helpful for the reader's understanding. Another possibility would be a small graphic that relates the development and the survey procedure to each other and illustrates the process.

Reply: We agree with this review suggestion. The paragraph has been moved to the beginning of section 3 (line 86-95). We also added one paragraph to briefly describe the development process of this board game (p.3, line 96-104)

Q4. The abstract should also make clear that EET has been developed and consequently tested. The processuality and the interrelationships in this development should be clearly formulated. So far EET appears as a generally known quantity, here context and explanation for the reader is missing. One or two sentences explaining the basic advantages of games for learning and the development of EET would be very helpful.

Reply: the abstract has been revised (line 9-12).

Q5. The authors sometimes jump between the grammatical tenses, for example in chapter 4 when "were" is used in the first sentence and in the following you use "are". This should be adjusted throughout the article.

Reply: we have adjusted this sentence and others ones.

Q6.Isn't figure 4 actually 2 figures with the same content? Either both photos should be numbered or only one photo should be selected. Another possibility would be again the naming in a and b as in figures 1 and 2.

Reply: This has been revised, adding (a) and (b)

Reviewer 2 Report

The work is very interesting, as is the card game proposed for teaching chemistry.

A very interesting part of the game design is the process that has been followed until the elements and rules are obtained. The authors could have commented on the game design process, that is, how decisions were made on types of cards, content, etc. If the process was with a multidisciplinary team.

It would also be interesting to know the content of the pre and post tests that were passed to the students.

The authors do not comment on any future work. Perhaps the game could be taken to a digital medium and could also be evaluated (computer game, tablet, etc.) and an evaluation could be made with a larger number of students.

Author Response

Reviewer 2

Comments and Suggestions for Authors

The work is very interesting, as is the card game proposed for teaching chemistry.

A very interesting part of the game design is the process that has been followed until the elements and rules are obtained. The authors could have commented on the game design process, that is, how decisions were made on types of cards, content, etc. If the process was with a multidisciplinary team.

Reply: We have added a paragraph and a figure (page 3) to describe the design process.

It would also be interesting to know the content of the pre and post tests that were passed to the students.

Reply: the test was designed in Mandarin. we have provided some examples in the section of Methods, due to the limit of space.

The authors do not comment on any future work. Perhaps the game could be taken to a digital medium and could also be evaluated (computer game, tablet, etc.) and an evaluation could be made with a larger number of students.

Reply: this article aims to report the game design and field-test results. We do not plan to develop it to digital platform. Rather, we are doing research with a larger number of students. This has been revised (page 10, line 358-360).
